

# Seasonal dynamics and environmental drivers of tissue and mucus microbiomes in the staghorn coral *Acropora pulchra*

Therese C. Miller[1,2,3] and Bastian Bentlage[1]

[1] Marine Laboratory, University of Guam, Mangilao, Guam, USA
[2] Institute of Marine Science, University of Auckland, Auckland, New Zealand
[3] Cawthron Institute, Nelson, New Zealand

Corresponding author
Bastian Bentlage,
bentlageb@triton.uog.edu

## ABSTRACT

**Background:** Rainfall-induced coastal runoff represents an important environmental impact in near-shore coral reefs that may affect coral-associated bacterial microbiomes. Shifts in microbiome community composition and function can stress corals and ultimately cause mortality and reef declines. Impacts of environmental stress may be site specific and differ between coral microbiome compartments (*e.g.*, tissue *versus* mucus). Coastal runoff and associated water pollution represent a major stressor for near-shore reef-ecosystems in Guam, Micronesia.

**Methods:** *Acropora pulchra* colonies growing on the West Hagåtña reef flat in Guam were sampled over a period of 8 months spanning the 2021 wet and dry seasons. To examine bacterial microbiome diversity and composition, samples of *A. pulchra* tissue and mucus were collected during late April, early July, late September, and at the end of December. Samples were collected from populations in two different habitat zones, near the reef crest (farshore) and close to shore (nearshore). Seawater samples were collected during the same time period to evaluate microbiome dynamics of the waters surrounding coral colonies. Tissue, mucus, and seawater microbiomes were characterized using 16S DNA metabarcoding in conjunction with Illumina sequencing. In addition, water samples were collected to determine fecal indicator bacteria (FIB) concentrations as an indicator of water pollution. Water temperatures were recorded using data loggers and precipitation data obtained from a nearby rain gauge. The correlation structure of environmental parameters (temperature and rainfall), FIB concentrations, and *A. pulchra* microbiome diversity was evaluated using a structural equation model. Beta diversity analyses were used to investigate spatio-temporal trends of microbiome composition.

**Results:** *Acropora pulchra* microbiome diversity differed between tissues and mucus, with mucus microbiome diversity being similar to the surrounding seawater. Rainfall and associated fluctuations of FIB concentrations were correlated with changes in tissue and mucus microbiomes, indicating their role as drivers of *A. pulchra* microbiome diversity. *A. pulchra* tissue microbiome composition remained relatively stable throughout dry and wet seasons; tissues were dominated by *Endozoicomonadaceae*, coral endosymbionts and putative indicators of coral health. In nearshore *A. pulchra* tissue microbiomes, *Simkaniaceae*, putative obligate coral endosymbionts, were more abundant than in *A. pulchra* colonies growing near the reef crest (farshore). *A. pulchra* mucus microbiomes were more diverse during the wet season than the dry season, a distinction that was also associated with drastic

shifts in microbiome composition. This study highlights the seasonal dynamics of coral microbiomes and demonstrates that microbiome diversity and composition may differ between coral tissues and the surface mucus layer.

## INTRODUCTION

Coral reefs are largely restricted to shallow waters that are strongly affected by environmental conditions that may vary across small spatial scales such as sea surface temperatures, water flow, and tidal ranges (*Guilcher, 1988*). Shallow near-shore coral reefs are also susceptible to anthropogenic impacts such as pollution and nutrient runoff (*Hughes, 1994*). Pollution by sewage, wastewater, and coastal runoff is a significant source of environmental stress that negatively impacts coral reefs by disrupting coral microbiome communities and their interactions with their coral host (*Wooldridge & Done, 2009*). Eutrophication by nitrogen and phosphorous has been shown to exacerbate coral bleaching (the loss of endosymbiotic Symbiodiniaceae algae in corals) and increase the prevalence and severity of disease (*Vega Thurber et al., 2014*). In Guam's near-shore reefs, *Redding et al. (2013)* found a positive correlation of sewage-derived nitrogen with severity of coral disease and identified that precipitation was correlated with the concentration of nitrogen, indicating that precipitation and pollution are closely linked. In addition to being a source of eutrophication, sewage and runoff carry bacteria into near-shore coral reef ecosystems (*Sutherland et al., 2011*; *Haapkylä et al., 2011*), potentially disrupting coral-associated bacterial microbiomes (*Maher, Epstein & Vega Thurber, 2022*). For example, bacterial microbiome diversity was shown to be elevated in corals exposed to sewage and municipal wastewater (*Ziegler et al., 2016*).

Bacteria are integral parts of the coral holobiont and are involved in nutrient cycling and the mitigation of pathogens (*Maher, Epstein & Vega Thurber, 2022*; *Bourne, Morrow & Webster, 2016*). Some bacterial taxa, such as *Endozoicomonas* (*Endozoicomonadaceae*, Gammaproteobacteria), represent important endosymbionts associated with coral health (*Neave et al., 2017*). *Endozoicomonas* plays a central role in dimethylsulfoniopropionate (DMSP) degradation leading to the release of dimethylsulfide (DMS) which is important for corals coping with thermal stress (*Chiou et al., 2023*). Loss of *Endozoicomonadaceae* has been linked with deterioration of coral health (*Meyer, Paul & Teplitski, 2014*). Conversely, some bacteria, most prominently species of *Vibrio*, are known to negatively impact coral health, inhibiting photosynthesis of endosymbiotic Symbiodiniaceae (*Banin et al., 2001*) and have been linked to coral tissue lesions (*e.g.*, *Wilson et al., 2013*).

Bacterial microbiomes may differ between compartments of the coral, including the skeleton, tissue, and the surface muco-polysaccharide layer (mucus hereafter) (*Glasl, Herndl & Frade, 2016*; *Hernandez-Agreda, Gates & Ainsworth, 2017*; *Peixoto et al., 2017*). The endolithic bacterial microbiome of the skeleton remains poorly studied and includes cyanobacteria, anoxygenic phototrophs, and anaerobic green sulfur bacteria that may

colonize layers deep in the aragonite skeleton (*Pernice et al., 2020*). The tissue layer is some 2–3 mm thick with several bacterial endosymbionts localized in cyst-like coral-associated microbial aggregates (CAMAs) (*Wada et al., 2019*). CAMAs may be intracellular and have been shown to be dominated by species of *Endozoicomonadaceae* and *Simkaniaceae* (Chlamydiota) (*Maire et al., 2023*). The mucus layer covers coral tissues and is in direct contact with the environment, acting as a first line of defense for corals against pathogens, playing an important role in disease mitigation through antibiotic activity (*Hernandez-Agreda, Gates & Ainsworth, 2017*; *Ritchie, 2006*). Disruption of mucus bacterial microbiomes has been linked to increases in *Vibrio* spp. and other pathogenic bacteria (*Glasl, Herndl & Frade, 2016*).

Coral bacterial microbiome composition and diversity may differ between habitats (*Camp et al., 2020*). The observation of such site-associated differences in coral microbiomes has led to the classification of some corals as microbiome conformers whose microbiomes adapt to the surrounding environment (*Voolstra & Ziegler, 2020*). Microbiome regulators, by contrast, maintain a consistent microbiome regardless of differences in external environment (*Voolstra & Ziegler, 2020*). For example, *Garren et al. (2009)* showed that *Porites cylindrica* transplanted to an environment impacted by fish farm effluent displayed elevated abundances of *Vibrio* spp. after five days of exposure to effluent, but microbiome composition reverted to its initial state three weeks after transplantation, indicating the capacity of *P. cylindrica* to regulate its microbiome. Microbiome conformers tend to accumulate and incorporate bacteria from the surrounding environment into their microbiomes. Several studies have found species of *Acropora* to be microbiome conformers whose microbiomes change following changes in the surrounding environment (*Greer et al., 2009*; *DeVantier et al., 2006*; *Ziegler et al., 2019*). Given these previous findings, we expected that the staghorn *Acropora* corals growing in Guam's near-shore reefs would display similar microbiome turnover when exposed to increased runoff and pollution during the wet season.

Staghorn *Acropora* spp., in particular *Acropora pulchra*, dominate Guam's reef flats, forming thickets whose extent has been reduced over the last decade by a combination of coral bleaching and extreme low tide exposure (*Raymundo et al., 2017*, *2019*, *2022*). Thus far, little is known about the bacterial microbiomes of Guam's *A. pulchra* populations and the impact seasonal runoff during the wet season may have on microbiome composition. We employed 16S rRNA metabarcoding to investigate the seasonal dynamics of *A. pulchra* bacterial microbiomes in Guam's West Hagåtña Bay. To identify possible differences in microbiome composition across coral compartments, microbiomes were characterized separately for both tissue and mucus. Specifically, the objectives of this study were to (1) characterize bacterial microbiome communities of coral tissue and mucus of *A. pulchra* from near-shore and far-shore zones of West Hagåtña Bay and (2) to elucidate the impacts of seasonal change on *A. pulchra* bacterial microbiomes. We found that the microbiomes of *A. pulchra* differed across compartments, with mucus microbiomes being more diverse than tissue microbiomes. The mucus microbiome behaved like a conformer, reflecting seasonal environmental changes, compared to tissue microbiomes that remained relatively

stable throughout the year, similar to a microbiome regulator. Nonetheless, both mucus and tissue microbiome diversity varied significantly between Guam's dry and wet seasons.

## MATERIALS AND METHODS

Portions of this text were previously published as part of a preprint (https://doi.org/10.1101/2023.09.07.556622).

### Field site

West Hagåtña Bay (Fig. 1A) lies on the west coast of the island of Guam, adjacent to the Hagåtña sewage treatment plant. The sewage outfall was renovated and repaired in 2008 to discharge 100 m further away from shore than it previously did; it currently sits 366 m beyond the reef line at a depth of 84 m (*Guam Waterworks Authority, 2019*). When tested for sewage-derived nitrogen isotopes (15N), soft corals in this bay contained twice as much 15N as those found in areas further removed from human impacts (*Redding et al., 2013*), indicating that this site is anthropogenically impacted. The land adjacent to West Hagåtña Bay is paved with a main road lined with stores, restaurants, and housing units. West Hagåtña Bay receives large influxes of runoff from land after rainfall events, increasing pollution above bacteriological standards, making waters unsafe for recreational activities (*e.g.*, *Howe, 2022*). West Hagåtña Bay contains three extensive staghorn coral thickets. The surface area of these thickets covers 121,439 m$^2$, 48.4% of which represents live coral cover. The dominant coral is *A. pulchra* and the estimated mean coral cover for this species is 23,992 m$^2$ (*Raymundo et al., 2022*). Given the shallow water depth (~0.5 to ~1 m depending on the tide), staghorn coral thickets in West Hagåtña Bay have a moderate to high vulnerability to aerial exposure from extreme low tides (*Heron et al., 2020*) which combined with elevated sea surface temperatures led to significant mortality of staghorn corals in 2014 and 2015 (*Raymundo et al., 2017*).

West Hagåtña Bay is influenced by a longshore current flowing from northeast to southwest (*Wolanski et al., 2003*) but currents within the bay have not been modeled in detail. However, *Fifer et al. (2021)* found significant differences in water flow speeds between the reef margin (far shore) and a site close to shore (near shore). These zones of differing water flow dynamics saw drastically different mortality rates during past coral bleaching and extreme low tide events, with the outer zone having <20% staghorn coral mortality while the inner zone saw >80% mortality (*Raymundo et al., 2017*). These differences in water flow and documented mortality were used to delineate the farshore (13.48219″ N, 144.7461″ E; 13.48222001″ N, 144.74458″ E; 13.48209998″ N, 144.74414″ E) and nearshore sites (13.48022997″ N, 144.7427″ E; 13.48004004″ N, 144.74262″ E; 13.48046802″ N, 144.7426″ E) targeted for this study. In each zone, three representative sites of *A. pulchra* stands were selected for sampling (Fig. 1B). These sites were chosen due to their high abundance of *A. pulchra* colonies that appeared healthy, with no visible bleaching, predation, or disease; coral colonies remained healthy throughout the duration of the sampling time period. At each site, three coral colonies were tagged to ensure repeated sampling of the same coral colonies. All coral collections for this study were made

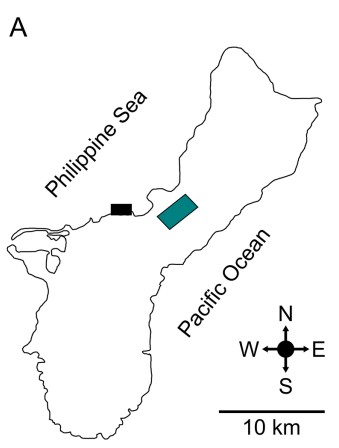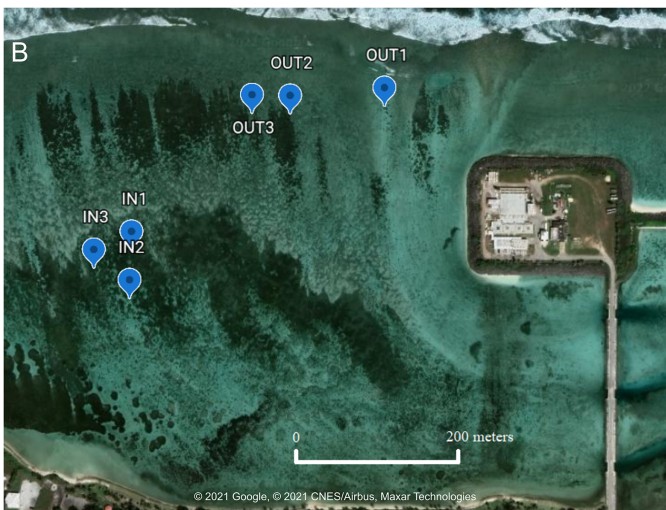

**Figure 1 Location of West Hagåtña Bay (black rectangle) in Guam (A) and sites targeted for sampling (B).** Sites in the inner (IN1-3) and outer (OUT1-3) zones where *Acropora pulchra* colonies were tagged for repeated sampling are indicated by blue markers. Latitude and Longitude for sampling locations are provided in Table S1. Green rectangle is the location of the Guam airport. Map data and image source credit: © 2021 Google, © 2021 CNES/Airbus, Maxar Technologies.

under a permit for the collection of coral issued by Guam's Department of Agriculture to the University of Guam Marine Laboratory.

## Environmental data

Six HOBO TidBit temperature loggers (Onset Corp., Bourne, MA, USA) were deployed at 1 m depth, one at each sampling site, to record seawater temperatures in 5 min intervals from April to December 2021, the duration of this study. Daily precipitation data were collected by the National Oceanic and Atmospheric Administration (NOAA) at the Guam International Airport, which lies approximately 6 km from the study site in the same drainage basin as Hagåtña Bay (*Taborosi et al., 2007*), and retrieved from the NOAA website (*National Oceanic and Atmospheric Administration, 2022*). During each of the four microbiome specimens collection events (see below), seawater samples were collected from each site using sterile 50 ml polypropylene centrifuge tubes; seawater samples were collected to measure fecal indicator bacteria (FIB) as a proxy for runoff and human impact. FIB concentrations (*Escherichia coli*, *Enterococcus*, and total coliform) were quantified by the Water and Environmental Research Institute (WERI) at the University of Guam. Separate 50 ml water samples were collected for determination of nitrate/nitrite ($NO_3^-/NO_2^-$) and ortho-phosphate ($PO_4^{3-}$) concentrations by WERI.

Temperatures were averaged by day within each zone (inner and outer), and daily averages were used in statistical analyses. Concentrations of FIB were averaged by month for each zone (inner and outer). Daily precipitation obtained from the *National Oceanic and Atmospheric Administration (2022)* was averaged by month for further analysis. Normality of each environmental dataset was estimated using a Shapiro-Wilk test. Two-way ANOVAs incorporating zones and months as factors were used to test for

significant differences in FIB concentrations between inner and outer zones as well as between months. A Kruskal-Wallis Chi$^2$ test was used to test for significant differences in rainfall per month. Seawater temperature differences between inner and outer zones as well as different months were tested for significance using Kruskal-Wallis Chi$^2$ tests on temperature.

## Microbiome sample collection

Samples were collected twice during Guam's dry season, at the end of April and beginning of July 2021, and twice during Guam's wet season, at the end of September and the end of December 2021. Coral nubbins (4.5–5 cm) were sampled from each of the nine tagged *A. pulchra* colonies in the outer zone and the nine tagged colonies in the inner zone. Sampling four times throughout the year yielded a total of 72 coral tissue and mucus samples each. The terminal end of each cut coral fragment was removed since new tissue layers at the growing tip may not be fully colonized by the coral microbiome. The mucus was sampled by exposing the cut coral nubbin to air and swabbing mucus from tissues using sterilized cotton swabs (*Lampert et al., 2008*). Coral nubbins were swabbed until tissues were visibly dry to ensure adequate mucus collection and that mucus contamination of tissue samples would be minimal. Coral nubbins, containing skeleton and tissue, and mucus samples were placed into separate Whirl-Pak sample bags (Filtration Group, Oakbrook, IL, USA). Samples were frozen in the field using liquid nitrogen and stored at −80 °C prior to DNA extraction and PCR.

To characterize the bacterial microbiome of the seawater in West Hagåtña Bay, 3 L of seawater were collected from the surface to a depth of 1 m during each collection event from each site. Seawater was stored in wide mouth plastic jars that had been sterilized for at least 30 min by soaking in 10% bleach solution prior to sample collection. Collected seawater samples were stored in coolers on ice for the approximately 30 min drive to the laboratory. Seawater was filtered using a 1.2 µm nylon filter (Sigma-Aldrich, St Louis, MO, USA) to collect bacteria for DNA extraction. Nylon filters were placed in Whirl-Pak sample bags and stored at −80 °C. Though 0.45 µm filters are commonly used to maximize recovery of bacteria, 1.2 µm nylon filters were chosen for this experiment to allow for filtration of large volumes of seawater; our attempts at using 0.45 µm filters resulted in the filters clogging due to the turbidity of collected seawater samples. While this choice may have reduced bacterial recovery, larger pore size filters are often used in environmental DNA (eDNA) studies as a trade-off to allow for filtration of large volumes of water to increase the odds of recovering rare taxa (*Zaiko et al., 2022*).

## DNA extraction and metabarcoding

DNA was extracted from coral tissue, mucus, and seawater samples using the DNeasy Powersoil kit (Qiagen, Hildenheim, Germany) following the manufacturer's protocol. Microbial DNA was extracted from mucus and seawater samples by placing cotton swab tips or membrane filters in bead beating tubes, followed by 30 s of homogenization. Tissue DNA was extracted by placing coral nubbins with tissue and skeleton into bead-beating tubes, followed by homogenization for 30 s, which resulted in disruption of tissues and the

surface layers of the skeleton, including coralites. *Pollock et al. (2018)* defined the coral skeleton as the aragonite layers not in direct contact with the coral tissue. The DNA extraction homogenate referred to as tissue by us contained the uppermost layer of skeleton, but not the deeper layers of the skeleton.

DNA concentrations were quantified using Qubit fluorometric quantification (Thermo Fisher Scientific, Waltham, MA, USA). Tissue DNA extracts were diluted to a concentration of 10 ng/µl; mucus and seawater extracts yielded around 1 ng/µl of DNA and were not further diluted. The V4 hypervariable region of 16S ribosomal DNA was amplified from each sample using universal bacterial primers primers 515F (*Walters et al., 2016*) and modified 806R (*Apprill et al., 2015*). The 30-µl PCR reactions included 3 µl of template DNA, PCR-grade water, 1x ExTaq buffer (Takara Bio, San Jose, CA, USA), 2.5 mM dNTPs, 10 µM forward and 10 µM reverse primers, and 0.75 U ExTaq DNA Polymerase (Takara Bio, San Jose, CA, USA). The thermocycler protocol comprised 30 cycles of initial denaturation at 95°C for 40 s, annealing at 58 °C for 2 min, and extension at 72 °C for 1 min; final elongation was performed at 72 °C for 5 min. A negative control using PCR-grade water instead of DNA as a template was included in every PCR run. All PCR products were checked on a 1% agarose gel stained with GelRed (Sigma-Aldrich, St Louis, MO, USA). PCR products that yielded one bright fluorescent band were considered successful. A subset of mucus and seawater samples that produced low yields from the initial PCR were re-amplified using additional PCR cycles. PCR products were purified using the GeneJet PCR Purification Kit (Thermo Fisher Scientific, Waltham, MA, USA) following the manufacturer's protocol and quantified using Qubit fluorometric quantification (Thermo Fisher Scientific, Waltham, MA, USA).

Purified PCR products were barcoded using indexes and MiSeq adapters synthesized by Macrogen (Seoul, Republic of Korea). Each indexing PCR reaction contained 2 µl of PCR product, 1 mM MiSeq adapter, PCR-grade water, 1x ExTaq buffer, 2.5 mM dNTPs, and 0.5 U ExTaq DNA polymerase per 20 µl reaction. Adapters were ligated to PCR products using five PCR cycles, including denaturation at 95 °C for 40 s, annealing at 59 °C for 2 min, and an extension at 72 °C for 1 min; a final elongation step at 72 °C for 7 min followed the five cycles. PCR products were pooled, resulting in two sequencing libraries that were purified using a GeneJet PCR Purification Kit (Thermo Fisher Scientific, Waltham, MA, USA), followed by resuspension of each library in 40 µl of elution buffer; each library pool contained a mixture of sampling timepoints to mitigate potential batch effects. 10 µl of each sequencing library were run through a 2% TBE agarose gel and the approximately 390 bp band representing the sequencing library was excised using a sterilized scalpel. 200 µl of nuclease-free water were added to the excised gel band and incubated overnight at 4 °C. DNA in the resulting solution was purified using the GeneJet PCR Purification Kit (Thermo Fisher Scientific, Waltham, MA, USA) and resuspended again in 20 µl of elution buffer. The size distribution of the purified sequencing library was verified using an Agilent 4150 TapeStation (Agilent, Santa Clara, CA, USA) with a D1000 ScreenTape assay. Libraries were sequenced at CD Genomics (Shirley, NY) using Illumina MiSeq (Illumina, San Diego, CA, USA) sequencing yielding 300 bp paired-end reads using v3 MiSeq chemistry.

## Sequence data quality control and taxonomic assignment

The R package DADA2 (*Callahan et al., 2016*) was used to remove primer sequences, truncate reads, calculate error rates, de-duplicate reads and infer amplicon sequence variants (ASVs) after merging of paired reads and removal of chimeras (87% of merged reads were non-chimeric). Non-chimeric ASVs were assigned a taxonomy from the Silva v138 dataset (*Glöckner et al., 2017*) using a naive Bayesian classifier (*Wang et al., 2007*) with a minimum bootstrap confidence of 50. The phyloseq package (*McMurdie & Holmes, 2013*) was used to remove ASVs whose taxonomy matched "Mitochondria," "Chloroplast", or "Eukaryota". MCMC.OTU (*Matz, 2016*) was used to remove ASVs representing <0.1% of count data and to identify putative outlier samples (those that had total counts falling below a z-score cutoff of −2.5). Two mucus samples with fewer than 1,000 reads (File S1) were discarded and not used in analyses. To assess adequacy of sequencing depth, a rarefaction curve was calculated which indicated that microbial diversity was largely captured across samples with fewer than 2,000 reads (Fig. S1). Given these results, no further normalization of samples was conducted to account for differential sequencing effort across samples (cf. *Amend et al., 2022*).

## Spatiotemporal variation of alpha diversity

Observed ASV diversity and Shannon diversity were calculated for each microbiome compartment (tissue, mucus, seawater) using the phyloseq package (*McMurdie & Holmes, 2013*). Evenness was calculated for each compartment by dividing the Shannon diversity index by the natural logarithm of observed diversity. Faith's Phylogenetic Diversity (PD; *Armstrong et al., 2021*) was calculated using the picante package in R (*Kembel et al., 2010*). Normality of diversity metrics for each compartment was tested using a Shapiro-Wilk test. For normally distributed diversity metrics, a two-way analysis of variance (ANOVA) was used to test for differences in alpha diversity across compartments, between months, and between inner and outer zones. Non-parametric analysis of variance (Kruskal-Wallis Chi$^2$ test) was used to compare alpha diversity when normality was rejected. A *post-hoc* Dunn's Test was then used for multiple comparisons.

## Structural equation modeling

To analyze the effects of environmental factors on microbiome diversity, structural equation models (SEMs) based on d-separation tests (*Shipley, 2009*) were created using the piecewise SEM package (*Lefcheck, 2016*). In particular, the relationships between precipitation, temperature, FIB concentrations, and microbiome diversity were evaluated. Linear models (LM) with Shannon diversity as the response variable and environmental variables, including precipitation, temperature, and FIB concentrations as the predictors formed the foundation of SEMs. The relations between FIB concentrations with precipitation and temperature were modeled using linear mixed effects models (LME). Precipitation and temperature were included as fixed factors and zone (inner and outer) was included as a random factor. Tissue and mucus microbiomes maintained a ratio of sample number to predictor variables (d) greater than 5, which is a widely used guideline for inclusion of predictor and response variables in structural equation models (*Lefcheck,*

*2016*). Given the low sample size for seawater microbiomes compared to predictor variables (d = 3.83), seawater was removed from consideration for final SEMs.

SEMs linked two exogenous variables (temperature and precipitation), three endogenous variables (FIB concentrations of *Enterococcus, E.coli*, and total coliform) and one response variable (Shannon diversity). SEMs tested for effects of (1) precipitation on concentrations of *Enterococcus*, total coliform, and *E. coli*, (2) temperature on concentrations of *Enterococcus*, total coliform, and *E. coli*, and (3) concentration of each FIB on microbiome diversity and the remaining two concentrations of FIBs (*e.g.*, concentration of total coliform on mucus microbiome diversity and concentrations of *E. coli* and *Enterococcus*). Testing for collinearity among FIB concentrations required inclusion of correlated error structures between each FIB. Separate SEMs that included the aforementioned tests were constructed for each compartment of the bacterial microbiome to further explore potential drivers of microbiome diversity.

### Beta diversity and differentially abundant taxa

PERMANOVA tests using the Adonis2 function in the vegan package (*Oksanen et al., 2022*) were run with 1,000 permutations to test for beta diversity differences between microbiome compartments, sampling months, and inner and outer zones based on weighted UniFrac distances; UniFrac distances were visualized using a Principal Coordinate Analysis (PCoA). Weighted UniFrac distances were used in these analyses because this metric integrates abundance information for ASVs in distance calculations (*Lozupone & Knight, 2005*). To test for significant differences and quantify similarity among sample sets, analysis of similarities (ANOSIM), as implemented in the phyloseq package (*McMurdie & Holmes, 2013*), was used to compare wet and dry season microbiomes as well as inner and outer zones of coral tissue, mucus, and seawater. *Post hoc* pairwise PERMANOVAs were used to identify possible differences between individual months or compartments.

The R package ANCOMBC (*Lin & Peddada, 2020*) was used to identify bacterial taxa that were differentially abundant between compartments. Considering that the mucus represents the interface in direct contact with both seawater and coral tissue, comparisons were made between mucus and seawater microbiomes as well as between mucus and tissue microbiomes. Taxa identified as differentially abundant with a false discovery rate <0.05 were considered statistically significant.

## RESULTS

### Environmental data

Water temperatures ranged from roughly 27 °C to 35 °C (Fig. S2). On average, the inner zone was warmer than the outer zone (Fig. S2). During the hottest part of the year, from June to August, there was on average a roughly 0.5 °C difference between inner and outer zones. However, the temperature difference observed between the two zones was not significant ($X^2$ = 3.181, df = 1, $p$ = 0.075). Water temperature did differ significantly when comparing across months throughout the year ($X^2$ = 275.51, df = 8, $p$ < 0.001). The amount

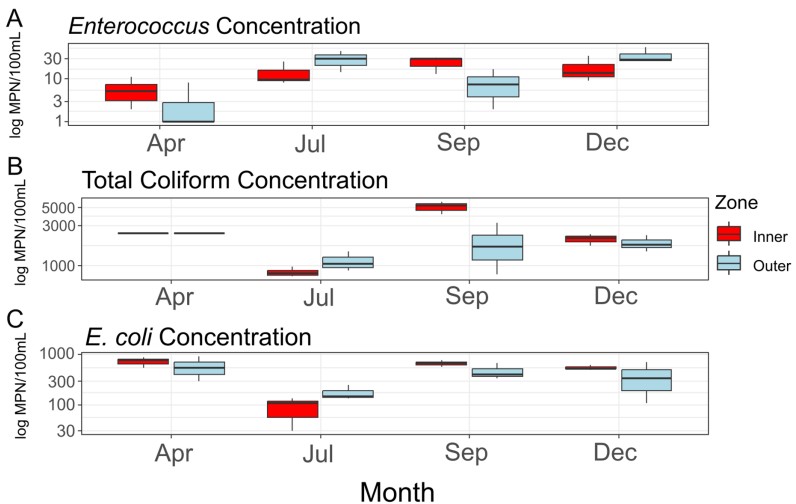

**Figure 2 Concentrations of fecal indicator bacteria (FIB) contained in water samples collected during the course of the study.** Concentrations (log$_{10}$ scaled) of fecal indicator bacteria (FIB) from each sampling time point. (A) Concentrations of *Enterococcus*; (B) concentrations of total coliform bacteria (total coliform concentrations for the month of April provided as >2,419.6 MPN/100 mL); (C) concentrations of *E. coli*.               

of precipitation significantly increased by the third sampling timepoint in September ($X^2$ = 12.39, df = 3, $p$ = 0.006; Fig. S3), as is typical of Guam's wet season.

Average concentrations for each FIB (Fig. 2) did not differ significantly between zones (*Enterococcus*: F = 0.216, df = 1, $p$ = 0.667; total coliform: F = 0.47, df = 1, $p$ = 0.531; *E. coli*: F = 0.258, df = 1, $p$ = 0.638), between sampling months (*Enterococcus*: F = 3.717, df = 1, $p$ = 0.126; total coliform: F = 0.01, df = 1, $p$ = 0.927; *E. coli*: F = 0.068, df = 1, $p$ = 0.807), or when taking the interactions of zones and months into account (*Enterococcus*: F = 0.300, df = 1, $p$ = 0.613; total coliform: F = 0.11, df = 1, $p$ = 0.756; *E. coli*: F = 0.025, df = 1, $p$ = 0.882). While not statistically significant, total coliform counts appeared elevated in the inner zone compared to the outer zone (Fig. 2B). No significant amounts of nitrite/nitrate or orthophosphate above a threshold of 0.01 mg/L were detected.

## Microbiome diversity

Throughout the duration of this study, the *A. pulchra* colonies sampled remained healthy without any visible disease lesions. Bleaching and algal overgrowth of *A. pulchra* tips due to low tide exposure was observed in September; these parts of colonies were not sampled. Raw reads of 16S metabarcoded data were deposited in NCBI GenBank's Sequence Read Archive (SRA; https://www.ncbi.nlm.nih.gov/sra) (Table S1). Average sequencing depth across samples after processing was 81,778 reads (File S1). Initially, 17,384 ASVs were identified; 1,355 (0.08%) of these were identified as eukaryotic and removed. After removal of low abundance ASVs (present in less than 0.1% of samples) and outliers, 321 ASVs were retained for downstream analysis. Sequence tags for each ASV and each sample, including average number of sequence tags within each compartment, are provided in File S1. After filtering, we retained a total of 292 ASVs for tissue, 312 ASVs for mucus, and 268 ASVs for

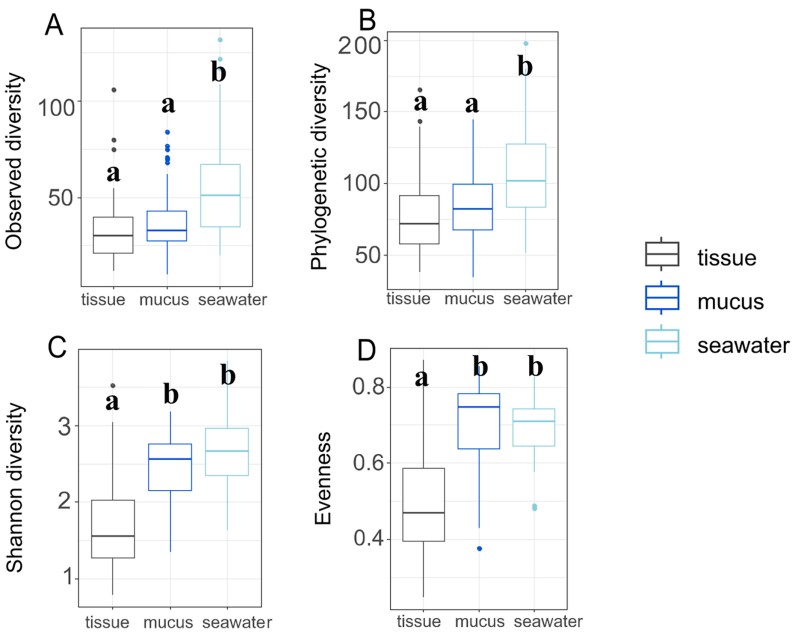

**Figure 3 Microbiome diversity and evenness of coral tissues, mucus, and seawater.** Boxplots depict minimum, first and third quartile, maximum, and median for observed diversity (A), phylogenetic diversity (B), Shannon diversity (C), and evenness (D). Statistical groups (a and b) were identified using *post-hoc* Dunn's tests.

seawater samples. Observed and phylogenetic ASV diversity was highest in seawater samples, with both tissue and mucus microbiomes being less diverse (Figs. 3A and 3B).

By contrast, Shannon diversity and evenness of mucus microbiomes was similar to seawater sample diversity and evenness and were higher than for tissue samples (Figs. 3C and 3D). Shannon diversity was not significantly different between inner and outer zones ($p = 0.322$) but we found significant variation across months ($p = 0.017$) (Table S2). Most interaction terms (month * compartment, zone * compartment, and month * zone * compartment), however, were significant in explaining variation in Shannon diversity ($p < 0.001$ for each) (Table S2). Zone ($p = 0.596$) or month alone ($p = 0.570$) did not explain the observed variation but all interaction terms including compartment were significant in explaining variation in evenness (Table S3). While variation of phylogenetic diversity of microbiomes was not explained by differences between zones alone ($p = 0.374$), phylogenetic diversity was affected by month ($p < 0.001$), compartment ($p < 0.001$), and all interaction terms of zone, month, and compartment (zone * month: $p < 0.001$; compartment * zone: $p = 0.002$; compartment * month: $p < 0.001$; compartment * zone * month: $p < 0.001$) (Table S4).

Interestingly, Shannon diversity was neither significantly different across zones ($p = 0.893$) nor months ($p = 0.106$) for tissue microbiomes (Table S2; Fig. S4) but the interactions of zone and month was significant for tissue microbiome phylogenetic diversity ($p = 0.020$) (Table S4) and tissue microbiome evenness ($p = 0.034$) (Table S3). Zone, month, or their interaction were significant factors influencing mucus microbiome diversity and evenness, by contrast (Tables S2–S4; Fig. S5); month was the only variable

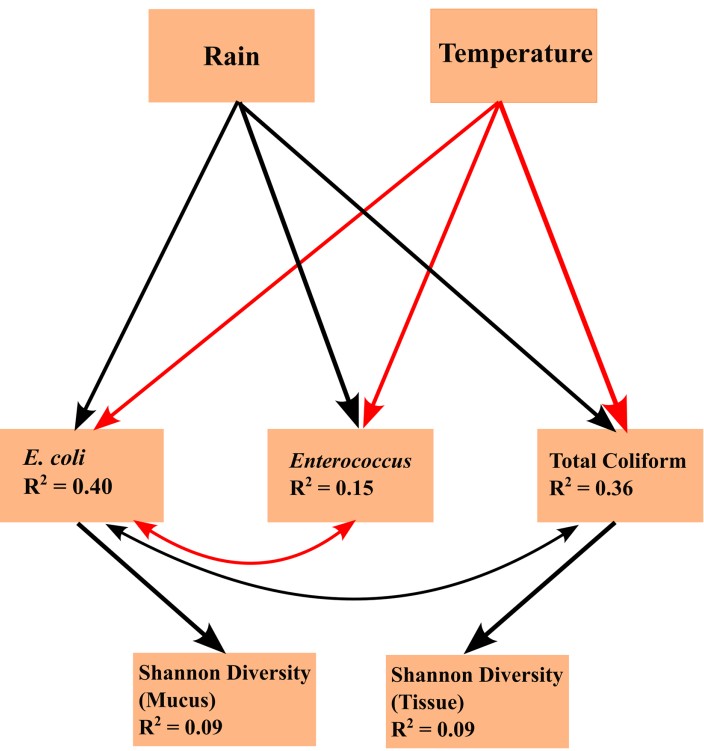

**Figure 4 Structural equation model (SEM) explaining the relationships between environmental parameters, fecal indicator bacteria (FIB) concentrations, and microbiome diversity.** The SEM shows the impacts of exogenous variables (rain and temperature) on endogenous variables (concentrations of *E. coli*, Enterococcus, and total coliform concentrations) and response variables (Shannon diversity of tissue and mucus). $R^2$ values are given for each endogenous and response variable. Arrows indicate statistically significant ($p < 0.05$) linkages. Red arrows indicate a negative correlation; black arrows indicate a positive correlation.

associated with seawater microbiome diversity, showing significant differences in phylogenetic diversity ($p = 0.012$) (Table S4; Fig. S6).

## Environmental predictors of microbiome diversity

Linear mixed effects model analyses for SEMs found the following statistically significant ($p < 0.05$) linkages: (1) an effect of temperature on FIB concentrations (total coliform, *E. coli*, and *Enterococcus*), (2) an effect of precipitation on FIB concentrations, (3) *E. coli* concentration is subject to common sources of variation with total coliform concentration and *Enterococcus* concentration, (4) an effect of total coliform concentration on tissue microbiome diversity; and (5) an effect of *E. coli* concentration on mucus microbiome diversity (Fig. 4). Temperature and FIB concentrations were negatively correlated, as were *E. coli* and *Enterococcus* concentrations (red arrows; Fig. 4), while all other correlations were positive (black arrows; Fig. 4). Total coliform concentration was a significant predictor of tissue microbiome diversity ($p = 0.044$, $R^2 = 0.09$) while *E. coli* concentration was a significant predictor for mucus microbiome diversity ($p = 0.019$, $R^2 = 0.09$). Both of these predictors were weakly associated with their responses, as indicated by the low $R^2$ values. Though inclusion of a correlated error structure between precipitation and mucus

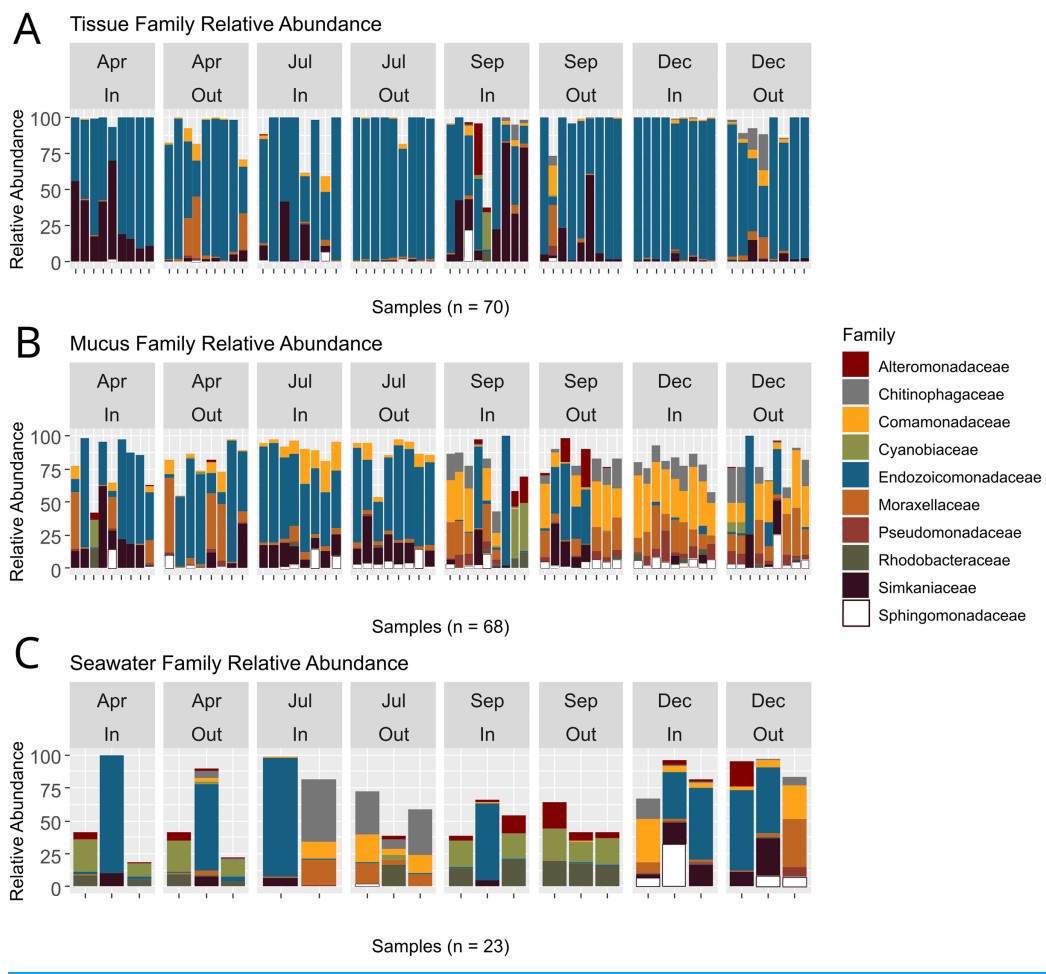

**Figure 5 Relative abundances of bacterial taxa identified in coral tissue, mucus, and seawater microbiomes.** The ten most abundant bacterial families found in the coral tissue (A), coral mucus (B), and seawater (C) are shown. Samples were separated by months and zones in which they were collected. April and July fell in Guam's dry season, and September and December fell in Guam's wet season.

microbial diversity was not found to be statistically significant ($p = 0.089$), and precipitation was not hypothesized to directly influence microbial diversity, inclusion of a correlated error structure greatly improved the model's goodness-of-fit ($p = 0.032$ to $p = 0.637$).

Arrows shown in the SEMs represent relationships supported by Fisher's C statistics (tissue: $C = 5.837$, df = 4, $p = 0.212$; mucus: $C = 0.901$, df = 2, $p = 0.637$). The fit of the model to the data could not be rejected by a global goodness-of-fit test ($C = 1.972$, df = 4, $p = 0.741$), indicating that no important pathways between variables in the SEM were excluded.

### Microbiome community composition

ASVs associated with the phylum Proteobacteria displayed the overall highest relative abundance, followed by Verrucomicrobiota and Bacteroidota (Table S5). Among

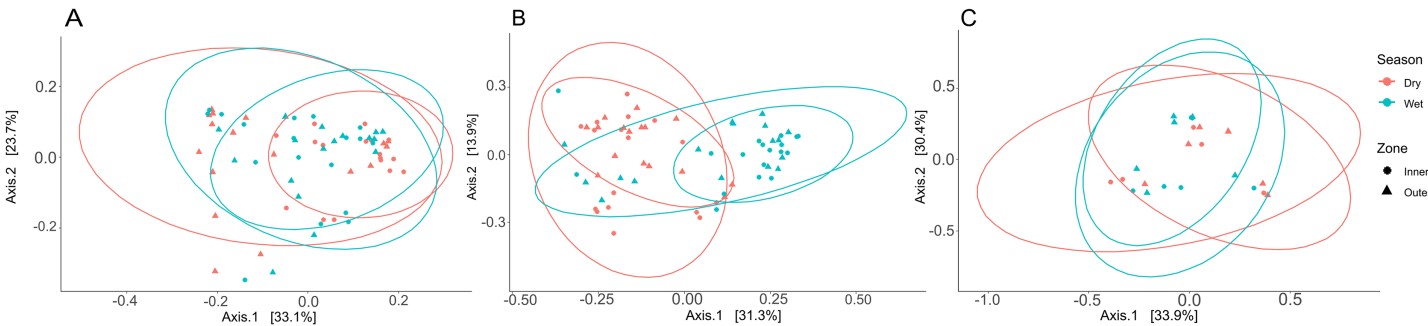

**Figure 6 Principal coordinate analysis (PCoA) plots to illustrate weighted unifrac distances of tissue (A), mucus (B), and seawater (C) microbiome communities.** The contribution of each axis to overall variation in the dataset is provided in percent. Colors indicate season, and shapes indicate reef zones from which samples were collected.

Proteobacteria, the family *Endozoicomonadaceae* was most abundant, especially in tissue microbiomes (Table S6; Fig. 5). In tissue microbiomes, *Endozoicomonadaceae* dominated all months and zones (Fig. 5A). While *Endozoicomonadaceae* were dominant in the mucus in April and July during the dry season, their abundance and prevalence dropped sharply in September and December during the wet season (Fig. 5B). *Simkaniaceae*, *Moraxellaceae* and *Comamonadaceae* were the other most abundant families in tissue and mucus microbiomes (Figs. 5A and 5B). While *Simkaniaceae* were more abundant in tissues compared to the mucus, *Comamonadaceae* and *Moraxellaceae* were more abundant and prevalent in the mucus compared to tissues (Table S6, Figs. 5A and 5B). *Endozoicomonadaceae* were also found in abundance in seawater samples, in addition to *Cyanobiaceae*, *Rhodobacteraceae*, and *Chitinophagaceae* (Table S6 and Fig. 5C).

PERMANOVA showed significant differences between months, but not between inner and outer zones (Table S7). Some interactions were significant (zone $^*$ month, compartment $^*$ month), but not all (compartment $^*$ zone, compartment $^*$ zone $^*$ month). Within tissue, there was a significant difference in beta diversity between inner and outer zones, as well as an interaction effect of zones and months, but not between months alone. For the mucus, there was no significant difference between inner and outer zones, or between the interaction of zones and months, but months differed significantly (Table S7). In the seawater, no statistically significant difference was seen by month, zone, or their interaction (Table S7). Pairwise PERMANOVAs (Table S8) revealed significant differences in beta diversity between each of the different compartments. Overall, significant differences in beta diversity were also seen between months: April *versus* December, July *versus* September, and July *versus* December. No significant differences were observed within tissue microbiomes when comparing sampling months. By contrast, mucus beta diversity differed across all months except for the comparison of September *versus* December. Seawater samples saw significant differences between September and December.

Three PCoA plots based on weighted UniFrac distances were constructed to visualize differences between zones and seasons for coral tissue, mucus, and seawater microbiomes (Fig. 6); ANOSIM was used to test for significant dissimilarities of microbiomes between

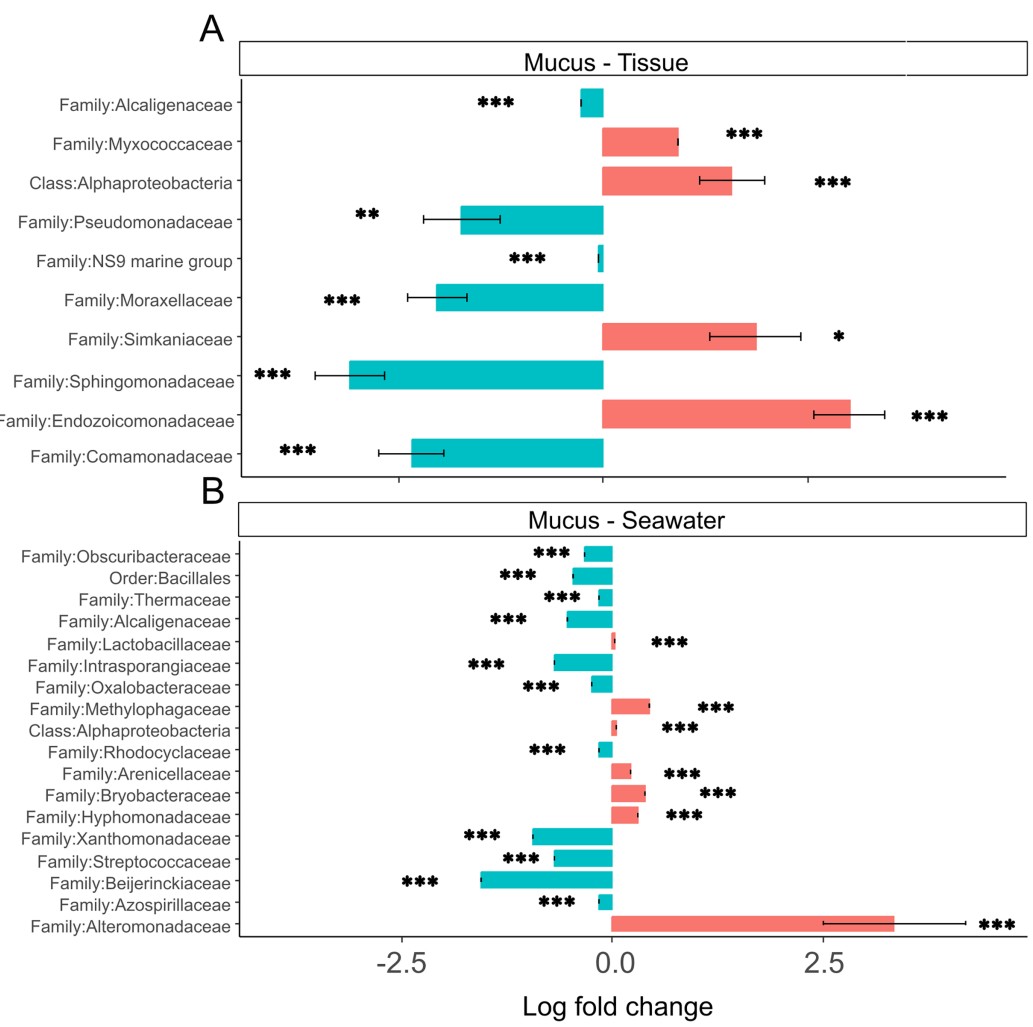

**Figure 7 (A and B) Differential abundance of bacterial taxa in microbiomes.** Bacterial taxa that were differentially abundant in comparisons of mucus *versus* tissue and mucus *versus* seawater microbiomes are shown. Asterisks indicate significance levels ($^*p \leq 0.05$; $^{**}p \leq 0.01$; $^{***}p \leq 0.001$).

zones and seasons (Table S9). Tissue microbiomes (Fig. 6A) differed between inner and outer zones during the dry season (R = 0.352; $p = 0.002$); wet and dry season tissue microbiomes showed a high degree of overlap but showed some significant differences between dry and wet seasons in the inner zone (R = 0.161; $p = 0.002$) and the outer zone (R = 0.169; p = 0.010). Mucus microbiomes (Fig. 6B) were differentiated by season (R = 0.315; $p < 0.001$) overall and for comparisons between wet and dry seasons within the inner zone (R = 0.309; $p = 0.005$) and outer zone (R = 0.417; $p < 0.001$). No statistically significant differences were found between seasons and zones for seawater samples (Fig. 6C).

## Compartment-specific microbiome differences

Tissue microbiomes were characterized by Alphaproteobacteria, *Endozoicomonadaceae*, *Simkaniaceae*, and *Myxococcaceae* that were significantly more abundant in tissue

microbiomes compared to the mucus (Fig. 7A). In the mucus, *Sphingomonadaceae*, *Comamonadaceae*, *Moraxellaceae*, and *Pseudomonadaceae* were significantly more abundant than in tissues (Fig. 7A). When comparing the mucus microbiome to seawater, the most striking difference was the increased abundance of *Alteromonadaceae* in seawater compared to the mucus (Fig. 7B). The most pronounced difference across compartments were the high abundances of *Endozoicomonadaceae* and *Simkaniaceae* in tissues, *Sphingomonadaceae* in the mucus, and *Alteromonadaceae* in the seawater (Fig. 7).

## DISCUSSION

Bacteria are an essential component of the coral symbiotic community that ensures homeostasis of the holobiont (*Boilard et al., 2020*). Environmental change and bacterial pollution may lead to disruptions of the coral bacterial microbiome, which can leave coral vulnerable to stress and disease (*Haapkylä et al., 2011*). Rainfall patterns during our study period reflected distinct wet and dry seasons with relatively low precipitation during the first two sampling time points in April and at the beginning of July that drastically increased by September, the third sampling time point (Fig. S3). While tissue microbiome taxonomic composition remained relatively stable throughout the year (Fig. 5A), mucus microbiome taxonomic composition showed a distinct and significant shift between the dry (April and July) and wet (September and December) seasons (Fig. 5B; Table S9). Contrary to the findings from some previous studies (*e.g.*, *Ziegler et al., 2019*), but in agreement with others (*Ziegler et al., 2016*; *Osman et al., 2020*), seawater microbiomes were most diverse (Fig. 3) and differed across all time points, albeit sample sizes were rather limited. West Hagåtña Bay is impacted by runoff which may explain high and variable seawater microbial diversity.

### Potential drivers of microbiome diversity

Consistent with rainfall patterns and resulting runoff into West Hagåtña Bay, *E. coli* and total coliform concentrations increased from July to September and remained high throughout December (Figs. 2B and 2C); high concentrations of coliform bacteria, including *E. coli* (Figs. 2B and 2C), in April were the likely result of a rain event the day prior to sampling. *Enterococcus* concentrations were elevated from July until the end of the wet season in December (Fig. 2A). While enterococci are used as indicators of sewage pollution similar to coliform bacteria, *Enterococcus* species may grow in different environments making them more general indicators of non-point source pollution from reservoirs, including soils in addition to sewage (*Rothenheber & Jones, 2018*). Considering that culverts drain residential runoff directly into the inner zone of West Hagåtña Bay near our sampling sites, it is not surprising that FIB concentrations were elevated in the inner zone compared to the outer zone in September during the height of the wet season (Fig. 2B); culverts and the Fonte River to the west of our study site are the sole drainage of surface stormwater runoff in the area, all of which terminate in West Hagåtña Bay.

We did not find any statistically significant differences in FIB concentrations across months or zones, but this could have been the result of low sampling effort. However, we found rainfall to be positively associated with FIB concentrations, in particular coliform

and *E. coli* concentrations (Fig. 4) that in turn were identified as weak but significant predictors of tissue and mucus microbiome diversity (Fig. 4). By contrast, water temperatures were negatively correlated with FIB concentrations. Overall, the SEM (Fig. 4) identified seasonal rainfall patterns and associated FIB concentrations as drivers of microbiome diversity. Variations in both tissue and mucus microbiome diversity were explained by month or the interaction of zone and month (Tables S5–S7). Zone alone was not able to explain differences in microbiome diversity, which suggests that the combination of increased runoff caused by seasonal rainfalls and distance from shore (inner *versus* outer zone) influenced microbiome diversity in *A. pulchra*.

## Microbiome composition through space and time

Previous studies compared coral microbiomes across different compartments (*Pollock et al., 2018*; *Marchioro et al., 2020*; *Sweet, Croquer & Bythell, 2011*), microhabitats (*Camp et al., 2020*; *Fifer et al., 2022*), or examined microbiome shifts over time (*Dunphy et al., 2019*; *Chu & Vollmer, 2016*; *Sweet, Croquer & Bythell, 2010*). In this study, we tracked the microbiome of *A. pulchra*'s tissue and mucus over time, comparing two habitats, near-shore inner and far-shore outer zones of West Hagåtña Bay. *A. pulchra* has previously been shown to bleach more severely in the inner zone compared to its conspecifics in the outer zone (*Raymundo et al., 2017*), likely caused by a combination of water temperature and flow differences (*Fifer et al., 2021*). Previous studies have shown that disease and bleaching may impact bacterial microbiome diversity and composition (*e.g.*, *Boilard et al., 2020*). We uncovered high relative abundances of *Simkaniaceae* and *Endozoicomonadaceae* in the tissues of inner zone *A. pulchra* (Fig. 5A), taxa that were characteristic of tissue microbiomes (Fig. 7A). Of all sequences assigned to *Endozoicomonadaceae* found in coral tissue samples by us, 54% originated from outer zone samples while 46% came from corals in the inner zone. *Endozoicomonadaceae* and *Simkaniaceae* have been shown to co-occur in corals, with *Simkaniaceae* being dominant in juvenile corals (*Bernasconi et al., 2019*). Members of *Simkaniaceae* have been commonly identified in coral microbiomes but their role remains largely unknown (*Ziegler et al., 2019*). However, recent research found that *Simkaniaceae* and *Endozoicomonadaceae* are located in adjacent CAMAs, potentially acting together as an important energy source for their coral host (*Maire et al., 2023*). Given these considerations, the high relative abundance of *Simkaniaceae* may be an acclimation to the environment of the inner zone of West Hagåtña Bay in *A. pulchra*.

While recent years have seen increasing research on the role of *Endozoicomonadaceae* on coral holobiont function (*Tandon et al., 2020*), the potential importance of *Simkaniaceae* for coral holobiont health has only been recognized recently (*Maire et al., 2023*). Interestingly, *Ziegler et al. (2019)* found an increase in relative abundance of *Simkaniaceae* in coral tissues at unimpacted sites compared to anthropogenically impacted sites. By contrast, we found that *Simkaniaceae* had higher relative abundances in near-shore, inner zone tissue microbiomes than far-shore, outer zone tissue microbiomes (Fig. 5A). Whole genome sequencing and annotation have shed light on the metabolic capacity of *Endozoicomonadaceae*, in particular species of *Endozoicomonas* (*e.g.*, *Neave*

*et al., 2017*; *Tandon et al., 2020*). Recently, *Endozoicomonas* was isolated from *A. pulchra* in Guam and its genome sequenced (*De La Vega, Shimpi & Bentlage, 2023*), laying the foundation for deeper insights into the metabolic repertoire of this endosymbiont. Genome sequences and isolates of *Endozoicomonas* species for experimental work have increased in recent years. Isolates and genome sequences of *Simkaniaceae* species are lacking but will be important for future comparative studies to understand their role in coral microbiome function.

While mucus bacterial communities were dominated by *Endozoicomonadaceae* during April and July (Fig. 5B), similar to tissue microbiomes (Fig. 5A), relative abundances of *Endozoicomonadaceae* in the mucus declined dramatically during the wet season in September and December. Runoff during periods of high rainfall may increase total dissolved solids, particulate nutrients that are frequently elevated in Guam's wet season (*Guam Waterworks Authority, 2019*), which may have led to the growth of the abundant *Rhodobacteraceae* and *Cyanobiaceae* present in seawater in September (Fig. 5C). Several mucus samples included high relative abundances for these taxa as well (Fig. 5B), likely originating from seawater. By December, *Comamonadaceae*, *Moraxellaceae*, *Chitinophagaceae*, and *Pseudomonadaceae* were the most abundant mucus microbiome families with community compositions relatively homogeneous across samples (Fig. 5B), suggesting microbiome acclimation. While their function remains largely unknown, these taxa have been found in the microbiomes of healthy corals (*Chu & Vollmer, 2016*; *McKew et al., 2012*; *Vijay et al., 2021*) and may provide benefits for *A. pulchra* when exposed to increased runoff during the wet season given their overrepresentation in the mucus (Fig. 7A).

## Microbiome conformism and regulation

Generally, *Acropora* spp. are considered conformers whose microbiomes are susceptible to and reflect shifts in the external environment (*Ziegler et al., 2019*). However, some acroporids have shown the opposite. For example, *A. tenuis* was found to be highly susceptible to the influence of the external environment on its microbiome while the bacterial microbiome of *A. millepora* remained stable regardless of environmental changes (*Marchioro et al., 2020*). When challenged with a new environment following transplantation, *A. hyacinthus* microbiomes appeared to acclimate to new and stressful conditions (*Ziegler et al., 2017*). In *A. pulchra*, we found overlap between mucus and seawater bacterial communities, suggesting that the mucus conforms to its surrounding environment. Tissue microbiomes, on the other hand, were relatively distinct and stable through time with community shifts largely restricted to changes in relative abundances of the dominant taxa *Endozoicomonadaceae* and *Simkaniaceae* (Fig. 5A). Consistent with previous studies (*Marchioro et al., 2020*; *Sweet, Croquer & Bythell, 2011*; *Amy, Weber Laura & Santoro Alyson, 2016*), we also observed tissue microbiomes to be less diverse compared to mucus and seawater bacterial communities (Fig. 3).

Bulk microbiomes are often extracted from combined tissue and mucus samples with the explicit aim of characterizing microbiomes of the entire coral holobiont (*e.g.*, *Pootakham et al., 2021*). Considering that environmental influences are not uniform across

the different microbiome compartments, examining microbiomes of the different layers of the coral holobiont separately has the potential to provide important insights into coral holobiont responses to environmental impacts. To address this issue, we characterized tissue and mucus microbiomes separately, albeit our tissue samples contained the top layers of the underlying skeleton. In *A. pulchra*, we found that the mucus acts as a microbiome conformer while the stability and relatively low diversity of tissue microbiomes suggests at least some degree of regulation. *A. pulchra* tissues were dominated by *Endozoicomonadaceae*, which are considered essential coral endosymbionts that produce antimicrobial compounds (*Rua et al., 2014*) and are positively correlated with Symbiodiniaceae densities (*Marchioro et al., 2020*). However, the association of *Endozoicomonadaceae* with corals may not be purely mutualistic (*Pogoreutz & Ziegler, 2024*).

Consistent with the results presented here, colonies of *A. pulchra* growing near the reef crest on New Caledonian reefs were dominated by *Endozoicomonas* while those living in lagoons were dominated not only by *Simkaniaceae* but also *Moraxellaceae* (*Camp et al., 2020*). *Endozoicomonas* spp. may secrete excess acetate which *Simkania* spp. may use as an energy source, ultimately aiding coral metabolism (*Maire et al., 2023*). The increased abundance of *Simkaniaceae* in inner zone *A. pulchra* tissues but also the outer zone in September during the height of the wet season in our study may represent an acclimation response that aids *A. pulchra* in mitigating the impacts of elevated water temperatures and rainfall-driven coastal runoff. Mucus microbiomes underwent noticeable change early in the wet season, with relative abundances of *Endozoicomonadaceae* declining and other taxa of bacteria increasing in relative abundance (Fig. 5B). Interestingly, *Rhodobacteraceae* were highly abundant in seawater and some mucus bacterial communities in September (Figs. 5B and 5C). *Rhodobacteraceae* co-occured with *Cyanobiaceae*, both of which are commonly found together (*Deignan & McDougald, 2022*; *Botté et al., 2022*). *Rhodobacteraceae* is a family comprising opportunistic heterotrophic bacteria that rapidly grow in the presence of organic-rich matter, as may be supplied by terrestrial runoff (*McDevitt-Irwin et al., 2017*).

High relative abundance of *Rhodobacteraceae* has been associated with declining *Endozoicomonadaceae* abundance, suggesting negative coral health impacts (*Pootakham et al., 2019*). By December, *Rhodobacteraceae* abundances had declined in mucus microbiomes and seawater (Figs. 5B and 5C) with mucus microbial relative abundances dominated by *Comamonadaceae*, *Moraxellaceae*, *Chitinophagaceae*, and *Pseudomonadaceae*. *Pseudomonadaceae* have previously been described from the mucus of healthy corals and are known for their antibacterial, antiviral, and antifouling properties that allow for control of viruses and prevention of biofilm formation (*Vijay et al., 2021*), linking this taxon to potential roles in microbiome regulation. *Moraxellaceae*, such as *Psychrobacter* spp., have also been described from coral mucus samples (*McKew et al., 2012*) and possess genes associated with carbon and nitrogen metabolism that may confer the ability to utilize organic compounds found in the mucus (*Badhai, Ghosh & Das, 2016*). While their functional role is not well understood, *Comamonadaceae* has been reported

from healthy corals and macroalgae found in reefs not impacted by environmental stressors (*Chu & Vollmer, 2016*; *Barott et al., 2011*; *Roder et al., 2014*).

Despite being in direct contact with the environment, mucus microbiome composition may in part be regulated by the coral host's physiology (*Glasl, Herndl & Frade, 2016*). *A. pulchra* mucus microbial relative abundances were highly variable in September and drastically differed from mucus sampled during earlier time points (Fig. 5B). By December, mucus microbiomes were relatively uniform but different from the microbiomes dominated largely by *Endozoicomonadaceae* in the dry season (Fig. 5B). This pattern suggests that mucus microbiomes were affected by environmental changes in the wet season, followed by possible coral host regulation of microbiome composition to a new state characterized by beneficial bacterial taxa. In addition, *Alteromonadaceae* were abundant in seawater during September (Fig. 5C) and were generally overrepresented in seawater compared to the mucus (Fig. 7B). *Alteromonadaceae* genera such as *Alteromonas* live freely in seawater and are associated with incorporation and possible translocation of nutrients to corals (*Ceh et al., 2013*). While we found no evidence of elevated concentrations of dissolved nitrogen or phosphate contrary to the results of FIB monitoring (Fig. 3), signatures of long-term eutrophication (*Redding et al., 2013*) as well as regular detection of elevated levels of total suspended solids (TSS) have been reported from West Hagåtña Bay (*Guam Waterworks Authority, 2019*).

## CONCLUSIONS

This study characterized the bacterial microbiome of coral tissue and mucus of the staghorn coral *Acropora pulchra* growing in nearshore (close to the shore) and farshore (close to the reef crest) habitats to examine shifts in microbiome diversity and composition between wet and dry seasons. Microbiome diversity in both coral tissue and mucus was influenced by seasonal rainfall patterns and associated bacterial pollution, highlighting the impact of runoff on coral microbiomes. However, *A. pulchra* tissue microbiome composition remained relatively stable across space and time despite *Acropora* species being considered microbiome conformers whose microbiomes closely resemble the surrounding environment. By contrast, *A. pulchra* mucus microbiomes were highly variable, with distinct differences in microbiome composition associated with the transition from the dry to the wet season. This study highlights the effects of coastal runoff and bacterial pollution on different compartments of the coral bacterial microbiome and identified seasonal rainfall patterns as a driver of coral microbiome diversity.

## ACKNOWLEDGEMENTS

We would like to thank all who contributed to this project, particularly Justin Berg, who assisted with field work and sampling. TCM would like to acknowledge Dr. Laurie Raymundo and Dr. Rebecca Vega Thurber for serving on her MS thesis committee on which the work presented here is based. We would like to thank Dr. Héloïse Rouzé for assisting with bioinformatic data analysis, particularly for her instructions on the use of DADA2 and phyloseq. We would further like to thank Dr. Brett Taylor for feedback on the construction of the structural equation model presented herein. We are indebted to three

anonymous reviewers whose critical feedback greatly improved an earlier version of this manuscript. Any opinions, findings, conclusions, or recommendations expressed in this material are those of the authors and do not necessarily reflect the views of the National Science Foundation.

### Funding

This work was supported by the National Science Foundation under cooperative agreement OIA-1946352. The funders had no role in study design, data collection and analysis, decision to publish, or preparation of the manuscript.

### Grant Disclosures

The following grant information was disclosed by the authors:
National Science Foundation: OIA-1946352.

### Competing Interests

The authors declare that they have no competing interests.

### Author Contributions

- Therese C. Miller conceived and designed the experiments, performed the experiments, analyzed the data, prepared figures and/or tables, authored or reviewed drafts of the article, and approved the final draft.
- Bastian Bentlage conceived and designed the experiments, performed the experiments, prepared figures and/or tables, authored or reviewed drafts of the article, and approved the final draft.

### Field Study Permissions

The following information was supplied relating to field study approvals (*i.e.*, approving body and any reference numbers):
Guam Department of Agriculture.

### DNA Deposition

The following information was supplied regarding the deposition of DNA sequences:
The raw sequence data from this project are available at GenBank: PRJNA1011454.

### Data Availability

The raw data and scripts used for data analyses are available in the Supplemental File.

### Supplemental Information

Supplemental information for this article can be found online at http://dx.doi.org/10.7717/peerj.17421#supplemental-information.

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
