# Peer review of "Seasonal dynamics and environmental drivers of tissue and mucus microbiomes in the staghorn coral Acropora pulchra"

_PeerJ, doi:10.7717/peerj.17421_

## Round 0.1 · original submission · Major Revisions

Your manuscript has been reviewed by three referees. Two of them suggested revision and one rejection. Please, pay special attention to Reviewers 1 & 3 and identify the suggestions you are able to address and those you cannot explain why.

Reviewer 1 ·

Basic reporting

no comment

Experimental design

no comment

Validity of the findings

no comment

Additional comments

General Assessment:
The overall innovation of the article is acceptable, and the methods and results are relatively reliable. However, in general, there is still a significant gap from publication for this article. At this stage, it is recommended to reject the manuscript or, after rejection, resubmit it. The main issues are twofold: 1) The writing logic of the article is chaotic, and the train of thought is unclear. 2) The discussion section lacks depth and mostly consists of a reiteration of the results.
Introduction
Overall, there is still a significant gap to reach the level required for publication in the introduction section. There are major issues with background explanation and logical coherence that require comprehensive revision. Specific suggestions are as follows:
Lines 54-61: Why introduce 15N here? It seems quite abrupt. How is 15N stable isotope related to the microbial composition the study focuses on? It's recommended to remove the relevant description in this paragraph. Instead, it would be better to directly address the topic of how terrestrial input or sewage might negatively impact corals by altering their microbial composition.
Lines 68-71: It would be beneficial to provide specific differences in the microbial composition between the skeleton, tissue, and mucus. For instance, what are the characteristic microbial communities in each component? What are their functions and ecological significance?
Lines 79-80: Compared to the previous section that discussed the microbial composition and changes in mucus, the introduction to the microbial composition of coral tissue here is rather brief. It's advisable to expand on this aspect in greater detail. In fact, the microbial communities within coral tissue are also dynamic under different temporal and spatial conditions, although they tend to be relatively more stable compared to those in mucus.
Lines 89-91: Are microbiome conformers more sensitive to environmental changes and more susceptible compared to microbiome regulators? Avoid just listing previous research findings; instead, provide your own analysis and propose reasonable hypotheses.
Lines 92-94: It seems somewhat odd to introduce the state of degradation in Acropora corals here. It is suggested to move this sentence to the next paragraph for better logical flow. Additionally, you mentioned that the degradation of Acropora corals is related to coral bleaching and extreme low tides. How does this relate to the factors this study is focusing on, such as different seasons, varying runoff, and different locations?
Lines 95-108: This section lacks background description of the Guam environment. For instance, what are the differences in seasonal runoff variations? How do the environmental conditions differ between nearshore and offshore coral reef areas? Is the disparity in environmental conditions between nearshore and offshore coral reef areas caused by variations in seasonal runoff?
Materials & Methods
Lines 115-116: The labeling for Fig. 2 in this sentence is incorrect.
Lines 147-149: How much mucus can be obtained from each branch? and how can you ensure that the mucus is collected thoroughly? Because collecting insufficient mucus may impact the microbial composition of coral tissues.
Results
Lines 268-274: Is the data represented as mean ± standard deviation? Are there no significant differences between any two groups, such as the two-fold difference in data between April and July in Figure 2b, but still no significant difference?
Lines 283-284: Specify the relevant p-values, as in the previous paragraph.
Lines 294-295: The result "with month alone also explaining microbiome evenness" is incorrect.
Lines 306- 308: The description in this sentence is inaccurate because E.coli and Enterococcus are also negatively correlated.
Discussion
The discussion section of this paper needs major revisions in order for it to be up to publishing standards. Please focus on revising this section to not only further explain your results, but to also include in-depth, supported reasoning for why you may have found such results and how your results fit into the current knowledge base that supports all the topics covered in this project. For example, from line 440 to line 455, the entire paragraph repeats the description of the results without any in-depth analysis. Such a discussion is meaningless. Also, the overall logic in the discussion section is quite confusing. It is also crucial that details are added to this paper to explain why the components of this project are important, not only to the benefit of individual corals but also for coral reefs as a whole.
Lines 409-411: The relative abundances of Endozoicomonadaceae was not high in the tissues of inner zone A. pulchra.
Lines 416-418: What is the function of Endozoicomonaceae? Among all the group samples, Endozoicomonaceae has the highest relative abundance. You cannot overlook its importance with just one sentence like this.
Lines 419-420: Simkaniaceae was not dominate throughout the year.
Lines 425-428: What is the function of Endozoicomonaceae? What impact does the reduction in the relative abundance of Endozoicomonaceae in mucus have? This sentence does not delve into a detailed discussion.

Reviewer 2 ·

Basic reporting

The study is thoroughly introduced with relevant literature and with a self-contained hypothesis. There are some small typos that I identified in my line-by-line comments. Most of the figures are presented clearly with the exception of Figure 6 which I think should be edited. For Figure 6, the authors state that month is represented in the figure, but that is not clear to me. In general, the figure is very busy and it is difficult to understand. Since it is established that mucus, tissue, and seawater have different microbial communities, each component should have its own individual PCA where month and zone can be highlighted more clearly.

Experimental design

The experimental questions are well-defined, and the analysis seems appropriate for the questions. But, clarification is needed on the taxonomic level (i.e., order, genus, ASV) used for beta-diversity and ANCOM analysis.

Validity of the findings

The statistical analysis supports the conclusions discussed.

Additional comments

Line 32 structural is misspelled
Line 114 add a sentence stating why these sites were picked within the zones
Line 159 Change DNE to DNeasy
Line 198 A closed parentheses is present without an open parentheses.
Line 199 I don’t know what non-bimeric means
Line 277 Supplemental Table 1, clarify that the sequences submitted to NCBI are not DADA2 reads, but raw unfiltered sequences.
Supplemental Fig 3-5, add y-axis label and any significance stats.

Reviewer 3 ·

Basic reporting

Article is very well-written, with minimal typos/grammatical errors, and professionally structured. However, the introduction stands to benefit from additional historical information about the study site and typical seasonal conditions or fluctuations. Additionally, I believe the introduction can be tweaked slightly to better reflect the study results within- a few portions do not pertain to the ultimate goals of the research. Some figures and tables (especially supplemental) are low-resolution and difficult to read or missing axis labels. For the literature cited, many findings run counter to previous research that is not cited in the manuscript, nor is it addressed in the writing. I think addressing those differences and discussing caveats and possible explanations for the discrepancies will strengthen their overall conclusions.

Experimental design

Research question is defined and the gap in knowledge identified. There are a few issues with experimental methods, especially the use of a 1.2µm filter for microbial characterization as this is not the field standard for marine microbial research, has the potential to confound the results by exclusion of a bacterial size fraction, nor is this choice explained or justified in the text. Authors will need to address this and discuss the caveats associated with this method in order to move forward with publication. Methods are also missing critical information associated with each aspect of the process: sampling methods/sampling areas, transportation, DNA extraction inputs, sequencing read depth, ASV numbers by compartment, read removal/manipulation, etc. This section needs a much higher attention to detail in order to demonstrate the potential for replication by the greater community.

Validity of the findings

Supporting data have been provided in a repository and statistical methods are appropriate and robust, but again the findings need to be discussed in connection with some of the more glaring caveats or gaps in methodological reporting, as well as the counter-findings to previous research testing many of the same hypotheses. I think this is critical for assessing the validity of the findings in the greater context of coral microbiome research as that is the main focus of the paper.

Additional comments

I have attached a PDF file with my in-line comments and suggestions for improvement where clarity and additional details are necessary. Molecular work with corals and host-associated microbes is always tricky, and I commend the authors on their work! With additional context, methodological details, and discussion of previous research and potential caveats, I believe this will be an interesting addition to the coral microbiome literature.

Annotated reviews are not available for download in order to protect the identity of reviewers who chose to remain anonymous.

---

## Round 0.2 · Minor Revisions

Your work has been reviewed by the same previous reviewers and are in agreement that your work need minor editorial revision. At this stage, I urge to do the last minor suggestions and resubmit.

Reviewer 2 ·

Basic reporting

The authors present the research clearly and cite appropriate references

Experimental design

The experimental design is sound.

Validity of the findings

The conclusions align with the analysis.

Additional comments

The authors addressed my concerns but there were some errors in the manuscript. See Below.

The authors state the temperature ranges up to 35C. But Supplemental Figure 2 does not show a temperature range up to 35.

Line 367 change to read 16S rRNA metabarcoded data

SupplementalFigure3.png is repeated twice.

Reviewer 3 ·

Basic reporting

As with the first draft, the revised manuscript is professional, clear, and self-contained. While the introduction is much improved from the first draft, it is still a bit hard to follow and still too long. The information and context provided is great, and the authors have clearly done their due diligence in researching and providing additional relevant literature- it just needs to be pared down a bit to be more clear and concise. Aside from a few typos, this is the only portion of the manuscript that I believe needs to be briefly revisited.

282- “Supplemental” has a typo. Also what “downsampling” mean? This is unclear.
446- unnecessary indent

Experimental design

Authors added additional details, context, and justification for methods that have improved the overall understanding/integrity of the results within- well done!

Validity of the findings

Data and analysis are robust and supported. The discussion is much improved and clear as to how the results relate the current literature and future investigations. Once again I applaud the authors on their revisions!

Additional comments

My first review was long and detailed, and the author's responses and reworking of the manuscript demonstrate a clear understanding of the work and its importance as part of the coral microbiome literature, especially in Guam! I have chosen "minor revisions" as I believe the introduction should be more concise, but otherwise would accept the manuscript. Well done!

---

## Round 0.3 · accepted · Accept

Your manuscript has been accepted.